# A National Portrait of Public Attitudes toward Opioid Use in the US: A Latent Class Analysis

**DOI:** 10.3390/ijerph20054455

**Published:** 2023-03-02

**Authors:** Suzan M. Walters, Weiwei Liu, Phoebe Lamuda, Jimi Huh, Russell Brewer, O’Dell Johnson, Ricky N. Bluthenthal, Bruce Taylor, John A. Schneider

**Affiliations:** 1Department of Epidemiology, School of Global Public Health, New York University, New York, NY 10003, USA; 2Center for Drug Use and HIV/HCV Research, New York, NY 10003, USA; 3Public Health Department, NORC at the University of Chicago, Chicago, IL 60603, USA; 4Department of Population and Public Health Sciences, Keck School of Medicine, University of Southern California, Los Angeles, CA 90033, USA; 5Department of Medicine, University of Chicago, Chicago, IL 60637, USA; 6Southern Public Health and Criminal Justice Research Center, University of Arkansas for Medical Sciences, Little Rock, AR 72205, USA

**Keywords:** public opinion, opioid use disorder, stigma, policy

## Abstract

Background: Opioid overdose rates have steadily been increasing in the United States (US) creating what is considered an overdose death crisis. The US has a mixture of public health and punitive policies aimed to address opioid use and the overdose crisis, yet little is known about public opinion relating to opioid use and policy support. Understanding the intersection of public opinion about opioid use disorder (OUD) and policy can be useful for developing interventions to address policy responses to overdose deaths. Methods: A national sample of cross-sectional data from the AmeriSpeak survey conducted from 27 February 2020 through 2 March 2020 was analyzed. Measures included attitudes toward OUD and policy beliefs. Latent class analysis, a person-centered approach, was used to identify groups of individuals endorsing similar stigma and policy beliefs. We then examined the relationship between the identified groups (i.e., classes) and key behavioral and demographic factors. Results: We identified three distinct groups: (1) “High Stigma/High Punitive Policy”, (2) “High Stigma/Mixed Public Health and Punitive Policy”, and (3) “Low Stigma/High Public Health Policy”. People with higher levels of education had reduced odds of being in the “High Stigma/High Punitive Policy” group. Conclusion: Public health policies are most effective in addressing OUD. We suggest targeting interventions toward the “High Stigma/Mixed Public Health and Punitive Policy” group since this group already displays some support for public health policies. Broader interventions, such as eliminating stigmatizing messaging in the media and redacting punitive policies, could reduce OUD stigma among all groups.

## 1. Introduction

Rates of opioid overdose deaths have been increasing since 1999 in the United States (US) [1,2,3,4]. Initially, the increase was attributed to prescription opioid pain relievers [1]. Yet, in 2010, opioid-related overdoses deaths were mainly attributed to heroin [2], in 2013 to fentanyl and its analogues [3], and since 2016 to polydrug use (e.g., cocaine, methamphetamine, and opioids) [4]. Furthermore, since the COVID-19 pandemic, overdose rates have continued to accelerate, and thus opioid and other drug related deaths continue to be a public health crisis [5,6]. Currently the main cause of overdose fatalities is illicitly manufactured fentanyl [4].

In an attempt to address the overdose death crisis, the US has instituted a mixture of punitive and public health policies focused on opioid use. Punitive policies include policies that criminalize people who use opioids nonmedically, which can result in arrest, incarceration, and a criminal record, all of which are associated with negative health outcomes [7,8,9,10]. Criminalizing drug use has resulted in unprecedented increases in the population of incarcerated persons in the US and health disparities, and these policies have disproportionally impacted minoritized communities [11,12,13]. People with a history of incarceration are disproportionately burdened with infectious diseases and stress-related illnesses. These impacts extend beyond the individual and also affect family members who are at increased risk for many negative health outcomes [14] and is reflected at the county level where higher incarceration rates are associated with increases in morbidity and mortality [15]. Furthermore, people who experience incarceration are at an increased risk of overdose, with the highest risk being immediately following release [16].

In contrast, public health policies view opioid use disorder (OUD) as a behavioral, social, and/or medical condition. Public health approaches may situate OUD in social, community, and structural forces that influence opioid use as well as health outcomes [17,18]. They also attempt to provide treatment and resources to people who use opioids; this can include access to medications for opioid use disorder (MOUD), such as methadone and buprenorphine, as well as harm reduction services. Both methadone and buprenorphine are associated with reduced overdose risk [19]. Harm reduction approaches do not necessarily focus on reducing drug use, rather they focus on minimizing drug use harms [20,21,22,23], and are associated with a reduction in infectious diseases as well as overdose [24,25,26,27]. An example of a public health policy that increases access to evidence-based treatment (e.g., MOUD) is Medicaid expansion. Medicaid expansion has been associated with decreased overdose deaths due to opioids [28] and has been deemed cost-effective [29]. MOUD availability in jails and prisons is important as well, as it is associated with reduced overdose [30]. An example of a harm reduction policy is increased access to naloxone, a drug that can reverse opioid overdose. States with policies that support naloxone distribution are associated with a decrease in fatal overdoses [31].

## 2. Theoretical Framework: Stigma

Stigma is a physical attribute, behavior, or a reputation viewed as outside the norm of society, which devalues a person or groups of people who live with stigmatized attributes [32]. Stigma is a social process where a difference is labeled, the difference is associated with a negative stereotype, there is a separation of “us” versus “them” (i.e., them as the stigmatized group), and results in status loss and discrimination [33]. Furthermore, stigma is a multidimensional social construct that operates at the individual, interpersonal, and institutional levels in interconnected ways [33,34]. Stigma operating at macro institutional levels is considered structural stigma.

### Structural Stigma: How Policies Reflect Societal Stigmas and How They Can Negatively Affect Individual Health Outcomes

Structural stigma is defined as “societal-level conditions, cultural norms, and institutional policies that constrain the opportunities, resources, and wellbeing of the stigmatized” [35]. Structural stigma is an understudied form of stigma; however, recent stigma research has engaged with these ideas, particularly by examining structural stigma in the form of policies [35]. Examples include policies that restrict gender and sexual minorities access to resources, such as prohibiting marriage and all benefits that come along with legalized unions [36], as well as policies that fail to protect sexual and gender minorities from bullying and discrimination [37], and how these create negative health outcomes for sexual and gender minorities. Researchers have also examined policies that restrict opportunities for people living with mental health illnesses [38,39]. Some research has explored effects of policies that criminalize diseases such as HIV [40], as well as healthcare and hospital policies stigmatizing persons who use drugs that serve as barriers to healthcare [41]. However, most research has not engaged specifically with structural stigma related to drug use policies, especially in relation to public support of stigmatizing policies; thus, more research is needed [36].

Drug use stigma has consistently been reflected in US policies since the 1800′s, when anti-opium laws were enacted [42]. Much research has focused on the negative effects of laws that criminalize drug use on communities, especially minoritized communities [12]. However, there is limited research on the public opinion of drug policies, and in particular opioid policies [43], which is an important area of study because public opinion can influence policy [44]. Past research has found that stigmatizing messages about opioid use lead to increased perceptions of persons who use opioids as dangerous and a desire to socially distance from them [45]. In addition, people who stigmatize and blame persons who use opioids are more likely to support punitive policies [46]. Conversely, persons who believe OUD is a medical condition [43] or have a personal connection to opioid use, either themselves or through a loved one, are more likely to favor public-health-oriented policies [47].

This study aimed to understand public opinions about opioid use and opioid policy. Specifically, we aimed to identify unique groups of people who support opioid-related policies, public health, or punitive approaches, with different levels of stigmatizing beliefs about opioid use and people who use opioids.

## 3. Methods

### 3.1. Study Sample

The analysis utilized cross-sectional data from the JCOIN AmeriSpeak^®^ Omnibus survey conducted from 27 February 2020 through 2 March 2020. AmeriSpeak is an ongoing probability-based panel of 35,000 households that conducts monthly surveys and is designed to be representative of the US household population (https://amerispeak.norc.org, accessed on 5 January 2023). More details about AmeriSpeak procedures can be found in previous publications [46,48].

In brief, US households were randomly selected using area probability and address-based sampling, with a known, nonzero probability of selection from NORC’s National Sampling Frame. AmeriSpeak uses mail, telephone, and face-to-face panel recruitment methods.

NORC’s sampling approach covers about 97% of the US household population [49]. AmeriSpeak is designed to meet the data quality standards of scientific and regulatory peer review and has the highest response rate of the available online probability-based panels [50].

A randomly selected group of eligible panel members, having already consented to respond to surveys, received an email that included a description of the stigma survey. The survey was offered in English and Spanish. Respondents who did not respond to the initial invitation were contacted multiple times by email and phone. Out of the 3915 invited, a total of 1036 respondents completed the survey with a response rate of 26.5%. Respondents were compensated 4 US dollars (USD) for their time. Respondents and nonrespondents were significantly different on a few demographic variables including sex, age, education, race/ethnicity, and region, in that respondents tended to be white, older, male, more educated, and from the Midwest region. Such differences were adjusted with nonresponse weights and incorporated into the final weights used to weigh the sample data.

Of the 1036 respondents, three had missing data on all latent class indicators and were removed. An additional 26 respondents had at least one missing covariate; removing these respondents reduced the analytic sample to 1,007. The sample data for this paper was weighted to US census benchmarks, taking into account selection probabilities (balanced by sex, age, education, race/ethnicity, and region) and nonresponse [51]. This study was approved by the Institutional Review Board of NORC at the University of Chicago.

### 3.2. Measures: Latent Class Indicators Measuring Public Attitudes toward Opioid Use Disorder

We first designed 39 items measuring attitudes toward OUD, including social stigma and policy beliefs toward OUD. These measures were adapted from past research [43,52,53]. We conducted exploratory factor analysis (EFA) and calculated Cronbach’s alphas for internal consistency of these measures to systematically reduce the number of indicators to include in the final latent class analysis (LCA) model. A final set of 24 items measuring seven constructs were included. The constructs were moral judgement of opioid use, social distancing from people who use opioids, perceived danger of people who use opioids, discrimination towards people who use opioids, support for public health-oriented policies, support for punitive policies, and support for evidence-based treatment. These measures reflect past research [43] and had acceptable internal consistency.

*Moral judgement of opioid use* was measured using five items (Cronbach’s α = 0.72). These items were adapted from NIDA’s fact sheets on general misperceptions of opioids [54] and refined with feedback from an advisory workgroup comprised of substance use researchers. Respondents rated their agreement with the following statements: “Most people who develop and/or struggle with opioid use disorder/addiction lack self-control”, “Misuse of opioids is a moral failing”, “A person struggling with opioid use disorder/addiction can quit using anytime if they choose”, “It is easy to find good opioid use disorder treatment”, and “Medication for opioid use disorder (e.g., methadone, buprenorphine, or naltrexone) is a hoax”. Responses were on a 5-point Likert-type scale ranging from “strongly disagree” to “strongly agree”. The mean of the five items was determined with higher values representing higher levels of moral judgement. A dichotomous indicator was then coded 1 for “more opioid moral judgement” if the mean of the five items was above the sample median and 0 for “less opioid moral judgement” if the mean was below the sample median, a method used in past studies using LCA [55,56].

*Social distancing from people who use opioids* was measured using two items (Cronbach’s α = 0.80). Respondents rated their agreement with the following statements: “I would be willing to have a person with a past history of opioid use disorder/addiction start working closely with me on a job” and “I would be comfortable having a person with past history of opioid use disorder/addiction marry into my family”. Responses were on a Likert-type scale ranging from “strongly disagree” to “strongly agree”. The mean of the two items was taken with higher values representing higher level of social distancing from people who use opioids. A dichotomous indicator was then coded 1 for “more social distancing” if the mean of the two items was above the sample median and 0 for “less social distancing” if the mean was below the sample median.

*Perceived danger of people who use opioids* was measured using four items (Cronbach’s α = 0.76). To assess perceived dangerousness, respondents rated their agreement with the following statements: “People with a current addiction to opioids/prescription pain medications are more dangerous than the general population”, “A person who is currently addicted to opioids/prescription pain medication cannot be trusted”, “I would be willing to have a person with a current opioid use disorder/addiction start working closely with me on a job”, and I would be comfortable having a person with a current opioid use disorder/addiction marry into my family”. Responses were on a Likert-type scale ranging from “strongly disagree” to “strongly agree”. The mean of the four items (the last two items were reverse-coded) was taken with higher values representing higher level of perceived danger of people who use opioids. A dichotomous indicator was then coded 1 for “more perceived danger of persons who use opioids” if the mean of the four items was above the sample median and 0 for “less perceived danger of persons who use opioids” if the mean was below the sample median.

*Discrimination towards people who use opioids* was measured using three items (Cronbach’s α = 0.56). To measure acceptance of discrimination, respondents rated their agreement with the following statements: “High schools and colleges should be allowed to dismiss or expel a person with an opioid use disorder/addiction”, “Employers should be allowed to deny employment to a person with a current opioid use disorder/addiction”, and “Landlords should be allowed to deny housing to a person with a current opioid use disorder/addiction”. Responses were on a Likert-type scale ranging from “strongly disagree” to “strongly agree”. The mean of the three items was taken with higher values representing a higher level of discrimination towards persons who use opioids. A dichotomous indicator was then coded 1 for “more discrimination towards persons who use opioids” if the mean of the three items was above the sample median and 0 for “less discrimination towards persons who use opioids” if the mean was below the sample median.

*Support for public health-oriented policies* was measured using four items (Cronbach’s α = 0.77). Respondents rated their agreement with the following statements: “I favor expanding Medicaid insurance benefits for low income families to provide coverage for treatment of opioid use disorders/addiction, including addiction to prescription pain medications”, “I favor making naloxone (also known as ‘Narcan’), a medication that can quickly reverse the effects of a person experiencing an opioid overdose widely available and affordable without a prescription”, “I favor increasing government spending to improve treatment of opioid use disorder/addiction”, and “Opioid use disorder is a medical condition like other chronic health conditions”. Responses were on a Likert-type scale ranging from “strongly disagree” to “strongly agree”. These public health-oriented policy measures were adapted from the work of Kennedy-Hendricks and colleagues [43]. The mean of the four items was taken with higher values representing higher level of support for public health-oriented policies. A dichotomous indicator was then coded 1 for “more support for public health policies” if the mean of the five items was above the sample median and 0 for “less support for public health policies” if the mean was below the sample median.

*Support for punitive policies* was measured using three items (Cronbach’s α = 0.76). Respondents rated their agreement with the following statements: “I believe that people in jail/prison is an effective way to improve the health of people with an opioid use disorder”, “I believe that incarceration/jail is an effective way to reduce the risk of overdosing for people with an opioid use disorder”, and “Jailing someone with an opioid use disorder for at least a few days will help them by reducing their risk for an overdose”. Responses were on a Likert-type scale ranging from “strongly disagree” to “strongly agree”. These punitive policy measures were adapted from the work of Kennedy-Hendricks and colleagues [43]. The mean of the three items was taken with higher values representing higher level of support for punitive policies. A dichotomous indicator was then coded 1 for “more support for punitive policies” if the average of the three items was above the sample median and 0 for “less support for punitive policies” if the mean was below the sample median.

*Support for evidence-based treatment* was measured using three items (Cronbach’s α = 0.93). Respondents rated their agreement with the following statements: “Individuals who are incarcerated with an opioid use disorder/addiction should get access to evidence-based treatment while incarcerated”, “Individuals who are on parole or probation with an opioid use disorder/addiction should get access to evidence-based treatment”, and “Evidence-based treatments for opioid use disorder can recover people from opioid addiction”. Responses were on a Likert-type scale ranging from “strongly disagree” to “strongly agree”. The mean of the three items was taken with higher values representing higher level of support for evidence-based treatment. A dichotomous indicator was then coded 1 for “more support for evidence-based treatment” if the average of the three items was above the sample median and 0 for “less support for evidence-based treatment” if the mean was below the sample median. These measures were adapted from the work of Kennedy-Hendricks and colleagues [43].

### 3.3. Covariates

We examined the relationship between class membership and the following covariates: gender (male or female), race and ethnicity (Black, Hispanic, white, other), education (no college; some college or more), marital status (married or living together; single), age, household income (above the state median; below the state median), and employment (employed; not working). We also included personal and family experiences with opioid use or overdose (yes; no) as well as personal and family history of criminal legal involvement (yes; no) as covariates, since past research has found these variables to be associated with opioid use stigma [48].

### 3.4. Analytical Model

We conducted a sensitivity analysis using latent profile analysis (LPA), where the continuous scales were used as class indicators, and results were similar to the LCA results presented in the paper. Using LCA, we identified groups of individuals endorsing similar stigma and policy beliefs (see Figure 1) [57]. LCA divides the population with similar response patterns into an unknown number of mutually exclusive and exhaustive subpopulations (or latent classes) [58]. LCA yielded information on how many subgroups of people who hold varying degrees of stigma and policy beliefs presented in the sample and what these subgroups look like. LCA models with different numbers of latent classes were estimated and compared based on substantive evaluation of the classes as well as fit statistics for non-nested models, such as Bayesian information criterion (BIC) and the Lo–Mendell–Rubin likelihood ratio test [59,60,61]. We also examined entropy values (ranges from 0–1), with a higher entropy indicating a better classification of individuals [61].

We examined the results in terms of specific latent classes of stigma and policy beliefs, the prevalence of these classes, and how covariates were related to the identified classes using a “three-step” approach developed by Vermunt (2010) and implemented in Mplus [62,63]. A latent class model was estimated (step 1), the most likely class variable from the posterior probability distributions was created (step 2), and the most likely latent class variable was regressed on predictor variables accounting for any misclassification (step 3). In this approach, classes are treated as latent rather than observed where uncertainty of class membership is incorporated in steps 2 and 3. Partial missing data class indicators were accounted for by using the widely accepted full information maximum likelihood (FIML) estimation method [64,65]. Results are weighted to ensure national representativeness. All analyses were conducted using Mplus version 7.3 (Los Angeles, CA, USA) [66].

## 4. Results

The weighted distribution of all study variables is presented in Table 1. Approximately 52% of respondents were female, 58% were married or living with a partner, 61% had some college education or more, and 60% reported a household income below the state median. Most respondents were white (64%), followed by Hispanic (16%), Black (12%), and other (8%). The average age was 48. About 40% reported having a family member who used opioids and/or overdosed and 13% reported personal use or overdose. In terms of criminal legal involvement, about 44% reported a family history of criminal legal involvement and 40% reported personal criminal legal involvement.

### 4.1. Latent Classes of Opioid Stigma and Policy Beliefs

Table 2 presents fit indices for models with different numbers of classes. Based on statistical criteria as well as substantive considerations, a 3-class model was selected (Log Likelihood (LL) = −4512.05 (23), BIC = 9183.14). Although the 4, 5, and 6-class solution presented lower BIC, the reduction from a 2-class solution to a 3-class solution is four times larger than that from a 3-class solution to a 4-class solution, indicating a diminishing return on BIC reduction. In addition, the Vuong–Lo–Mendell–Rubin adjusted likelihood ratio test (VLMR-LRT) p-value indicated a borderline significant model fit improvement comparing a 3-class model with a 2-class model when using the conventional alpha of 0.05 (*p*-value = 0.09). The p-value was not statistically significant when comparing a 4-class model with a 3-class model (*p*-value = 0.35). Based on the comparison of the VLMR-LRT test results and literature suggesting that BIC is a more reliable index than VLMR-LRT test [59], we ultimately decided on a 3-class with optimal balance between model fit and parsimony.

Figure 2 shows the three distinct classes identified based on opioid use stigma and policy beliefs. Just over one third (37%) of respondents in this sample belonged to a “High Stigma/High Punitive Policy” class. Individuals within this group showed a high probability of moral judgment of opioid use (0.71) and avoidance of persons who use opioids (0.80). They also showed a high probability of discrimination (0.56), perceiving persons who use opioids as dangerous (0.55), and had a higher probability of supporting punitive policies to address OUD (0.68). This class largely did not support a public health approach (0.08) or evidence-based treatment (0.08).

A second class, representing slightly over a third of respondents (37.6%) was identified as “High Stigma/Mixed Public Health and Punitive Policy”. This class had a high probability of discrimination (0.80) and perceiving persons who use opioids as dangerous (0.73). They also showed a moderate probability of opioid moral judgment (0.47) and avoiding persons who use opioids (0.44). However, unlike the “High Stigma/Punitive Policy” group, individuals in this group had a high probability of supporting public health approaches to address opioid dependency (0.66) and believing in evidence-based treatment for OUD (0.65). However, this group also had a fairly high probability of supporting punitive policies to address OUD (0.58).

A quarter (25.4%) of respondents belonged to a third class identified as “Low Stigma/High Public Health Policy”, where respondents showed low probability of opioid moral judgement (0.22) and discriminating (probability = 0.10) against persons who use opioids. They also showed the lowest likelihood of avoiding persons who use opioids (0.18) and perceiving persons who use opioids as dangerous (0.20). The probability of supporting a public health approach (0.89) and evidence-based treatment (0.80) were high while showing a low probability of supporting a punitive policy approach to responding to opioid use (0.12).

### 4.2. Associations between Individual Characteristics and Class Membership

Controlling for other individual characteristics, education is related to class membership (all covariates were added simultaneously; Table 3). Relative to the “Low Stigma/High Public Health Policy” class, individuals with higher levels of education were 65% less likely (AOR = 0.35; *p*-value = 0.01) to belong to the “High Stigma/High Punitive Policy” class than those who had no college education. Gender, age, race and ethnicity, marital status, employment, income, personal or family history of opioid use, and personal or family history of criminal legal involvement were not associated with “High Stigma/High Punitive Policy” class nor “High Stigma/Mixed Public Health and Punitive Policy” class membership, compared to the “Low Stigma/High Public Health Policy” class.

## 5. Discussion

Using a nationally representative sample, this study sought to identify groups of people based on their levels of opioid use stigma and policy beliefs. We identified three distinct groups with differences in stigma and policy beliefs: (1) a “High Stigma/High Punitive Policy” group, (2) a “High Stigma/Mixed Public Health and Punitive Policy” group; and (3) a “Low Stigma/High Public Health Policy” group. The first two groups were more common among respondents. Of concern is that only 25.4% of the sample was in the “Low Stigma/High Public Health Policy” group, suggesting the need for more work in addressing stigma towards opioids. This study expands on previous work that found that persons who had higher opioid use stigma endorsed greater discrimination towards persons who use opioids, lower support for public health policies, and higher support for punitive policies [43,46]. To our knowledge, this is the first study to use LCA in studying public stigma toward opioid use. Results can be used to better support people who use opioids via targeting interventions focusing on identified groups with the goal of decreasing stigma and enhancing support for constructive public health policies.

We found an association with education, where people with higher levels of education had reduced odds of being in the “High Stigma/High Punitive Policy” group membership, suggesting that higher education lends itself to more public-health-oriented beliefs. Surprisingly, we did not find personal or family opioid use or personal or family history of criminal legal involvement to be associated with any group membership at a statistically significant level. However, personal opioid use was close to significance, with lower odds of being in the “High Stigma/High Punitive Policy” class (*p* = 0.10), which would be expected. Conversely, family opioid use was positively associated with being in the “High Stigma/Mixed Public Health and Punitive Policy” group compared to the “Low Stigma/High Public Health Policy” at a near significance level (*p* = 0.12). It may be that family members hold stigmatizing views about their relatives who use opioids and thus have mixed public health and punitive policy beliefs. Past research has documented that family members stigmatize family members who use drugs [67] and that family members are major sources of dignity attacks experienced by people who use drugs [68]. These findings gesture to larger societal forces, such as social institutions (i.e., media, policies, education), that structure beliefs, as opposed to personal and interpersonal experiences that occur at the micro level. The education system may be a structural factor that buffers opioid use stigma while the media, a large societal institution, may exacerbate opioid use stigma [69].

The media has played a historical role in framing drug use, including opioid use, in negative, and often racist ways [70,71,72]. Researchers have argued that recent opioid use narratives have been more sympathetic than past drug use narratives, which were more punitive. Opioid messaging includes more support for public health-oriented approaches, such as support for drug treatment, towards persons who use opioids, rather than criminalization [72]. The media and public discourse shift from punitive beliefs to public health beliefs has been problematized as racist rhetoric where mainly people of color were depicted as having problematic drug use (e.g., with crack cocaine), targeted and then criminalized in the past [12,71]. Conversely, the opioid “crisis” has garnered greater sympathies, and has been depicted as affecting more white nonurban communities [70,73].

Of great importance is that media depictions such as these can translate into policy support. A recent study compared news articles from the New York Times and Washington Post found that the media medicalized opioid use for white people and in contrast villainized crack use for Black people. The authors then conducted a vignette experiment where respondents received a fictional news article with the type of drug, heroin or crack cocaine, randomly inserted. Respondents were then asked whether the person in the vignette should be charged with a crime or assigned to drug treatment. The authors found that people were more likely to support criminalization for Black persons regardless of the type of drug used and drug treatment for white persons [74]. Furthermore, policy research finds that people who hold more negative attitudes toward Black people (i.e., racial stigma) have decreased odds of supporting public health policies addressing opioid use [75]. This inequity seems to be occurring in drug courts as well, as mandated treatment is more likely to be assigned to white people than people of color [76,77].

This study did not find race or ethnicity to be significant; however, we controlled for individual race and ethnicity, not beliefs about people of different races and ethnicities. Structural racism is a main driver of health inequities for persons who use drugs [78,79]. Previous studies measured racial beliefs and racism [75], and it is possible that racist beliefs influence stigma beliefs and policy supports, therefore future research should explore this. Future research should examine support for public health versus punitive policies by race and ethnicity. Another area for study is exploring associations between racist beliefs by adding scales that measure racism. Research should incorporate measures related to racism, and how racism intersects with drug use stigma and policy beliefs.

Our findings suggest that a group that might be best suited for stigma reduction interventions is the “High Stigma/Mixed Public Health and Punitive Policy” group. They also were more likely to have a family history of opioid use, which could be why they support some public health policies. We suggest this group because they already support some public health policies; however, they also support punitive policies. It is possible that with a reduction in opioid use stigma this group may lean more heavily toward public-health-oriented policies [43]. One possible intervention could be an information campaign that frames drug use as a treatable medical condition [80]. Additional suggestions for decreasing opioid use stigma include replacing stigmatizing language (e.g., “addict”, “abuse”) with more person-centered language [80,81], framing public health messages about opioid use in continuum language (e.g., recognizing that drug use fluctuates and that people may use different amounts and/or no drugs at different points of their lives), and involving people with drug use histories in public health efforts [82], including through storytelling and sharing experiences related to drug use [83].

Despite a more sympathetic media narrative toward opioid use [70], stigma towards people who use opioids is still pervasive and is often reflected in language. A ten-year content analysis of print and television news stories found an increase in stigmatizing language (terms such as “addict” and “substance abuse”) from 2008–2009 to 2017–2018, indicating that the media may play an important role in contributing to opioid use stigma, and that this stigma may impact policy beliefs [84]. How opioid use is framed and reflected in language and images throughout society can influence the levels of stigma that people feel toward persons who use opioids [85]. This may be part of the reason why many people prefer to distance themselves and socially exclude persons who use opioids [86].

Similarly, policies criminalizing drug use can affect public opinion about opioids and create stigmatizing beliefs [43]. However, punitive policies are not effective. For example, a study exploring punitive policies (e.g., criminalization) versus public health policies (e.g., expanded treatment) related to women using drugs while pregnant found that states with punitive policies had greater odds of neonatal abstinence syndrome. The authors of this study conclude that punitive policies do not work as a deterrent for using drugs while pregnant, but rather serve as a catalyst for healthcare disengagement [87]. Some research argues that people hold beliefs not by reason, but rather by notions of morality that are cultural and learned. Although this may be true, another argument that seems to be more aligned with our findings, which adds to the aforementioned, is that morality intersects with racism and other isms [88] that are used to keep people down, in, and away [89].

The US should also examine how public health-oriented policies can be improved. This includes increasing access to drug treatment, improving treatment experiences and retention, and making treatment options equitable for all. Racial and ethnic disparities in drug treatment exist, with greater access among white people [90]. Persons of color have limited access to drug treatment while incarcerated [91], and are more frequently prescribed methadone, which is administered in highly-monitored and stigmatizing settings [92]. This contrasts with white people who are more frequently prescribed buprenorphine, a drug that can be self-administered without direct supervision [93,94,95,96]. Additionally, white people are more likely to receive mandatory treatment in drug courts while people of color are more likely to be sentenced to jail or prison [76].

This study is not without limitations. First, respondents self-reported their perceptions, which may be subject to recall and/or social desirability biases. However, the reliability and validity of self-reported behaviors such as drug use [97] and in criminal legal research [98] has been established in past research. Second, we did not have measures on racism and therefore were not able to include stigma towards a person’s race and/or ethnicity. Future research should explore the intersection of racial and drug use stigma. Third, this was a cross-sectional study, and we therefore cannot draw causal inferences compared to if we have had a longitudinal design. Fourth, this survey was conducted during the first wave of the COVID-19 pandemic [99] and during a time when the US experienced numerous Black Lives Matter protests in response to anti-Black racism, as well as increases in hate crimes toward people of color, including Asian people [100]. This specific time in history could have influenced public opinion and research should look longitudinally at public opinions to evaluate this. Fifth, the paper focused on a limited set of opioid-related policies. It is possible that if we had a broader set of measures of opioid-related policies we might have been able to add a broader set of recommendations in our paper. The survey items used in this paper were part of a brief survey mechanism used by AmeriSpeak as part of their Omnibus monthly survey. Therefore, the research team was limited in how many questions could be asked of the AmeriSpeak participants. Finally, while the sample has been weighted to represent the nation and nonresponse weights were adjusted using the observed variables, it is still possible that respondents and nonrespondents were different on unobserved variables.

## 6. Conclusions

This study identified three unique groups of people based on their levels of opioid use stigma and policy beliefs. We identified a “High Stigma/High Punitive Policy” group, “High Stigma/Mixed Public Health and Punitive Policy” group, and “Low Stigma/High Public Health Policy” group. More so, we find that people with higher levels of education had decreased odds of being in the “High Stigma/High Punitive Policy” group. We suggest targeting interventions toward the “High Stigma/Mixed Public Health and Punitive Policy” group since this group already displays some support for public health policies. Broader interventions, such as eliminating stigmatizing messaging in the media could reduce opioid use stigma among all groups. Future research should continue to explore the relationship between stigma and policy support.

## Figures and Tables

**Figure 1 ijerph-20-04455-f001:**
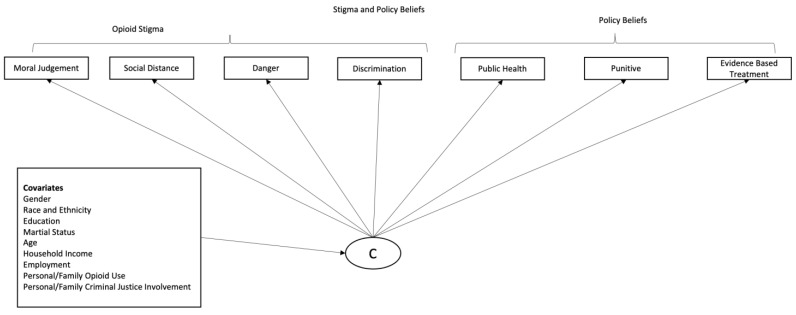
Groups of individuals endorsing similar stigma and policy beliefs.

**Figure 2 ijerph-20-04455-f002:**
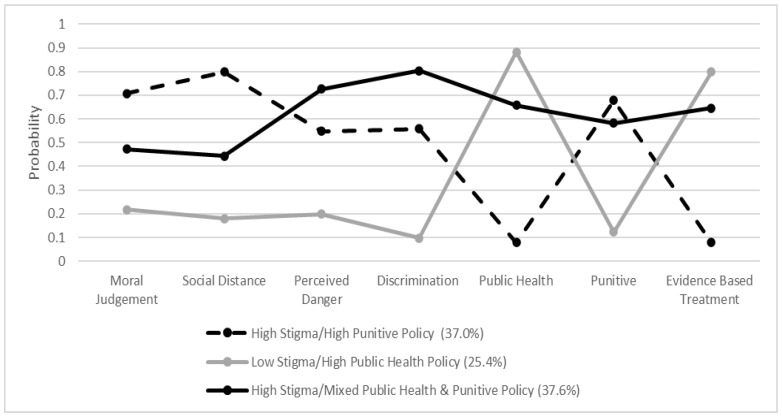
Latent Classes of Opioid Stigma and Policy Beliefs: results from the nationally representative AmeriSpeak survey, February 2020.

**Table 1 ijerph-20-04455-t001:** Weighted distribution of outcomes and covariates: data from the nationally representative AmeriSpeak survey, February 2020 (n = 1007).

	Frequency/Mean (Std. Dev)
**Class Indicators**	
Moral judgement of opioid use *	49.5%
Social distance from individuals who use opioids *	50.7%
Perceive individuals who use opioids as dangerous *	52.6%
Discrimination against individuals who use opioids (housing and employment) *	53.4%
Support public policies to cover OUD treatment and protect individuals who use opioids (i.e., public health) *	50.0%
Support public policies that criminalize opioid use (i.e., punitive) *	50.1%
Support for evidence-based treatment *	47.4%
**Covariates**	
**Gender**	
Male	48.3%
Female	51.7%
**Race/Ethnicity**	
White	63.8%
Hispanic	15.9%
Black	11.9%
Other	8.4%
**Education**	
No college	38.6%
Some college or more	61.4%
**Marital Status**	
Married or living together	58.3%
Single	41.7%
**Age**	48.1 (17.9)
**Household Income**	
Above state median	40.1%
Below state median	59.9%
**Employed**	54.5%
**Experiences with Opioid Use and/or Overdose**	
Personal opioid use or overdose	13.2%
Family opioid use or overdose	40.2%
**Criminal Legal Involvement**	
Personal criminal legal involvement	40.2%
Family criminal legal involvement	44.2%

* We report the frequency of the dichotomous indicator (e.g., more of) for the variables listed.

**Table 2 ijerph-20-04455-t002:** Latent class analysis: model fit statistics to determine stigma beliefs and policy support classes: data from the nationally representative Amerispeak survey, February 2020 (n = 1007; results weighted to ensure national representativeness).

Model	LL ^a^	No. of Parameters	BIC ^b^	VLMR-LRT ^c^
1 class	−4879.70	7	9807.81	N/A
2 class	−4585.84	15	9275.41	<0.01
3 class ^d^	−4512.05	23	9183.14	0.09
4 class	−4469.95	31	9154.25	0.35
5 class	−4429.38	39	9128.43	0.26
6 class	−4414.94	47	9154.86	0.52

^a^ Log likelihood; ^b^ Bayesian information criterion (lower value indicates a better fit); ^c^ Lo–Mendell–Rubin likelihood ratio test *p*-value (significant *p*-value suggests that a K-class model fits better than a model with one less class); ^d^ selected model.

**Table 3 ijerph-20-04455-t003:** Associations between individual characteristics and class membership *.

	“High Stigma/High Punitive Policy”		“High Stigma/Mixed Public Health and Punitive Policy”	
	AOR	*p*-Value	AOR	*p*-Value
Gender				
Male	1.11	0.74	0.76	0.41
Female	Ref		Ref	
Race/Ethnicity				
Black	1.82	0.16	1.01	0.98
Hispanic	1.49	0.39	0.94	0.92
Other	0.92	0.90	2.59	0.11
White	Ref		Ref	
Age	1.00	0.83	1.02	0.13
Marital Status				
Married or living together	0.87	0.67	1.21	0.62
Single	Ref		Ref	
Employed				
Yes	1.39	0.34	1.00	0.99
No	Ref		Ref	
Education				
Some college or more	**0.35**	**0.01**	0.48	0.06
No college	Ref		Ref	
Household income				
Below state median	0.65	0.17	0.99	0.99
Above state median	Ref		Ref	
Personal opioid use				
Yes	0.36	0.10	0.55	0.37
No	Ref		Ref	
Family opioid use				
Yes	1.01	0.98	1.86	0.12
No	Ref		Ref	
Personal criminal legal involvement				
Yes	0.80	0.74	1.91	0.35
No	Ref		Ref	
Family criminal legal involvement				
Yes	0.70	0.27	0.67	0.26
No	Ref		Ref	

Note: Bold font indicates statistical significance; * Low Stigma/High Public Health Policy class is the reference class.

## Data Availability

Data available on request due to restrictions.

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
