# Peer review of "A National Portrait of Public Attitudes toward Opioid Use in the US: A Latent Class Analysis"

_ijerph, 2023, doi:10.3390/ijerph20054455_

Round 1
Reviewer 1 Report
Thank you for the opportunity to review this interesting and timely manuscript about public attitudes on policies to address opioid use disorder (OUD). Some strengths of the study are its large sample size (N ~ 1000), its representative sampling strategy, and the creation and validation of 7 new scales measuring public policy and stigma attitudes related to OUD. The new scales show good internal consistency reliability (alphas > .70) and provide useful information about 3 different types of policies often seen in OUD-related legislation - punitive, public health/prevention, and treatment approaches. In and of itself, the grouping of disparate public policies into these 3 buckets is a meaningful contribution. The authors then extend that work by classifying survey respondents into 3 groups or constituencies based on their mix of attitudes toward these policy types, as well as their other attitudes toward people with OUD. The use of stigma as an overall theoretical framework is another strength of this study.
I have one major methodological concern about this otherwise excellent paper, and a few minor suggestions for improvement. My one major concern is the use of median splits on each of the 3 policy-related and 4 stigma-related variables. The dichotomous versions of each variable were then used to identify the 3 groups of respondents using latent class analysis. The problem is that median splits are notorious for distorting results, by forcing continuous variables into categories that may not be ecologically valid. This can artificially create the appearance of divisions or categories where they do not truly exist -- a major problem when categorization is the goal of the study. The best-practice solution would be to replace latent class analysis with latent profile analysis, which would handle each of the 7 scales as a continuous variable and would hopefully produce comparable results -- e.g., in Figure 2 the probabilities in the y-axis would be replaced with average scores on each scale.
An alternate way to address this problem would be to show that the variables are truly bimodal -- e.g., a strong grouping of participants at the high vs. low ends of each of the scales. Median splits are most problematic for normally distributed variables, so if these variables aren't normally distributed then maybe there is no problem. Or, the authors could perform some sensitivity analyses to show that if the variables were handled as continuous the results would still be similar. At the very least, the authors could provide a table of means and standard deviations to show that scores really were meaningfully different on each of the variables for groups on the high vs. low side of the median split -- i.e., that these differences are clinically meaningful and not simply a matter of a few tenths of a point. The main point is that the authors need to convince readers that the median splits didn't create artificial groups when the reality is on a continuum.
My other minor suggestions are as follows:
1. The Introduction draws a contrast between policies toward OUD that are punitive, and policies that take a public health approach. Public health vs. treatment approaches could be further differentiated here, as they are in the measures developed by the investigators. A public health approach, for instance, does not necessarily need to be based on a view of OUD as a medical problem -- it could be seen as a social or behavioral problem instead. Items on the public health scale were mainly focused on resource availability, although one of the items did assess whether the participant viewed OUD as a medical condition.
2. The summary of punitive policies in the 2nd paragraph focuses on different outcomes than the summary of pubic health polices in the 3rd paragraph. Punitive policies increase the number of people incarcerated, and they exacerbate health disparities; but do they reduce the total amount of opioid use or the overdose rate? The argument here would be stronger if the investigators were also to present evidence that the more punitive polices fail to achieve these important goals, which the public health policies are more effective in achieving. The only policy intervention that the investigators specifically tie to a reduction in the overdose death rate is naloxone distribution. Buprenorphine treatment also has evidence for reducing overdose deaths, and it seems like this is a consistent benchmark that could be applied to all of the policy choices reviewed. Incarceration is presented in the 3rd paragraph as a problem in itself, but those in favor of more punitive policies might argue that incarceration is not, per se, a problem as long as it achieves the goals of reduced opioid use and overdose deaths. I don't think it does, I'm just saying that the argument would be stronger if all policy options were compared to the same yardstick.
3. As noted above, the scales developed to measure attitudes toward people with OUD and policies related to OUD were innovative. I can see clear applications for this tool in future research. However, I wondered if the first scale -- "disregard of OUD as a medical condition" -- could be more clearly named "moral judgment." The term disregard made me think that people were disregarding OUD itself, in other words believing that OUD is unimportant or is not a problem that society needs to address. Perhaps the term should be disagreement that OUD is a medical condition. But even that label conveys an assumption that OUD really is a medical condition, and the problem is that people who score high on this scale do not share that assumption. The label "moral judgment" (or "moral model"?) might be more neutral in the sense that people who score high on this scale really do subscribe to an alternative moral model of opioid use, and they see people with OUD as being in need of judgment rather than in need of help.
4. The associations between policy attitudes and personal or family opioid use in Table 3 were very interesting. I was surprised to see that people in the "high stigma/mixed policy" group were actually more likely than those in the "low stigma/high public health policy" reference group to have (a) a family member who used opioids, and (b) personal justice system involvement. The authors address one of these in the Discussion but not the other. I wondered if having personal justice system involvement might predispose people to a more absolute view of right and wrong, or to prioritize justice over treatment considerations for OUD. In other words, it seems that those who have been on the wrong side of the law themselves are actually the most likely to hold others accountable to the letter of the law. It would be interesting to know the authors' thoughts on this finding, as well as the finding about family use.
5. A useful resource to consider for the Discussion and/or for further research might be Haidt's Moral Foundations Theory (J. Haidt, 2012, The Righteous Mind: Why Good People are Divided by Politics and Religion. New York: Pantheon). Haidt differentiates between an ethics of care and an ethics of fairness and authority. I can see these same divisions in the clusters of policy attitudes that were discovered in the current study. Consideration of other moral foundations from Haidt's theory might be helpful in understanding the mix of attitudes in the "high stigma/mixed policy" group -- for example, maybe a violation of ideas of group loyalty, or a "disgust" reaction tied to the sanctity/degradation moral foundation, is involved in this group's higher endorsement of stigma even though they also support more humane treatment-focused policies. Better understanding these underlying attitudes might provide further direction for designing media messages targeted to this specific group.
Reviewer 2 Report
The manuscript titled “A National Portrait of Public Attitudes Toward Opioid Use in the US: A Latent Class Analysis” uses a national household survey data and latent class analysis approach to identify population sub-groups with different attitudes towards opioid use. It is a well-written manuscript and contributes meaningfully to the public health literature on opioid use. I have a few comments:
i) The authors created a composite outcome for the 7 constructs by dichotomizing the average of the items based on the median. Although the authors cite previous papers which have used the same approach, I don’t think it’s a statistically sound approach. I think it somewhat distorts the prevalence of the three latent groups. For example, for the public health construct, if the majority of the sample had responded strongly agree to public health measures, the zero value for the binary construct would also capture who had responded agree for those items.
I would have preferred calculating factor scores using one of the methods and using the factor scores to construct the latent groups. But since the approach the authors used has been previously used, I think the authors can at least mention this in the limitation. Also, I think it would make the manuscript better if the authors could report the percentage of responses for each level for the 25 items used (as a supplement table).
ii) There are a few typos in the manuscript: Chronbach’s alpha (instead of Cronbach’s alpha), where (line 382; which should be “were”).
Reviewer 3 Report
This is an important topic, and the paper is well-written. My main concern is related to the choice of policies classified as public health-oriented and punitive policies. In this article, some of the policies seem to be specific for justice-involved persons, while others seem to be related to the general population with OUD, many of who may not be involved with the justice system. Justice involvement, type of substance used, and race/ethnicity of the person with OUD are deeply interconnected with stigma, and they have a more significant impact on outcomes for people of color. Please see below specific comments.
The introduction could be more developed, especially where the authors talk about mandated treatment. Are they referring to court-mandated treatment in place of jail time or mandated treatment before release? The authors should provide more context for this public health policy example.
In the method section, the authors say that some panel members selected to participate did not respond to the survey, but it is not clear how many. They only report the final number of respondents (1,036). Please include the response rate considering the total number of participants selected to participate. It would also be helpful to know whether the respondents differed (for example, in education and income) from the panel members selected to participate. The authors mention the use of non-response weights and cite a study from Canada without providing information on how the non-response rates were calculated. Please explain how non-response rates were used in the present study.
Related to the previous item, in the limitations section, please explain how non-response could influence your results/recommendations. For example, if people who decided to participate in the study were likely to have a profile that is more or less compatible with the classes identified, how would that affect your results?
The item “I believe that making drug treatment mandatory is an effective way to help people with an opioid use disorder” (line 201) is not clear. Is this referring to mandatory treatment for the overall population? I don’t believe that population mandatory treatment is a proposed public health policy and that many people will rightly oppose it. The authors should exclude this item from the “support for public health-oriented policies” construct.
Punitive policy items used in this study seem to be only related to justice-involved individuals. However, punitive policies also affect people with OUD outside of the justice system or before they enter the justice system. See some examples: https://jamanetwork.com/journals/jamanetworkopen/fullarticle/2755304
https://ps.psychiatryonline.org/doi/10.1176/appi.ps.201600056
The authors should explain why they focused on a limited set of policies. Further, in the discussion, they should talk about how limiting the scope of policies would have repercussions for their conclusions and recommendations.
Minor comment: In Table one, the authors have “Disregard OUD as a medical condition” – 49.5%. I am assuming that this is the percentage of respondents with high disregard (i.e., above the average for that component). Is that correct? If so, please specify that in the table.
Round 2
Reviewer 1 Report
Thank you for your time and attention to detail in addressing my comments on the previous version of this manuscript.
I want to particularly acknowledge your careful, scholarly response to my question about latent profile vs. latent class analysis. You put in an extreme amount of new analytic work for what ultimately amounted to a footnote, yet I think this was an important step and I greatly appreciated your systematic response. After seeing the supplementary analyses, I agree with retaining the latent class analysis approach in the manuscript: The graph is more interpretable that way, and the overall pattern of results is the same.
I also appreciated the additional context and clarifications that you added to the text, which I think strengthen this manuscript. (I will look up the book Dying of Whiteness). I have no further comments or concerns about this manuscript. Thank you for the dialogue and for your work to improve this paper.
Reviewer 3 Report
-